

# GFADE: generalized feature adaptation and discrimination enhancement for deepfake detection

ZhiYong Tian[1] and Junkai Yi[2]

[1] School of Automation, Beijing Information Science and Technology University, Beijing, China
[2] Key Laboratory of Modern Measurement and Control Technology Ministry of Education, Beijing Information Science and Technology University, Beijing, China

## ABSTRACT

With the rapid advancement of deep generative techniques, such as generative adversarial networks (GANs), the creation of realistic fake images and videos has become increasingly accessible, raising significant security and privacy concerns. Although existing deepfake detection methods perform well within a single dataset, they often experience substantial performance degradation when applied across datasets or manipulation types. To address this challenge, we propose a novel deepfake detection framework that combines multiple loss functions and the MixStyle technique. By integrating Cross-Entropy Loss, ArcFace loss, and Focal Loss, our model enhances its discriminative power to better handle complex forgery characteristics and effectively mitigate data imbalance. Additionally, the MixStyle technique introduces diverse visual styles during training, further improving the model's generalization across different datasets and manipulation scenarios. Experimental results demonstrate that our method achieves superior detection accuracy across a range of cross-dataset and cross-manipulation tests, significantly improving model robustness and generalizability.

# INTRODUCTION

The emergence of realistic image generation through generative adversarial networks (GANs) has raised significant security concerns, particularly regarding the potential for malicious bio-informatics, where human portraits can be easily substituted. Face manipulation, especially in the form of deepfake technology, has become a growing concern in our digital society. Deepfake, a term derived from "deep learning" and "fake", refers to using artificial intelligence to synthesize highly realistic but entirely fabricated images, videos, and audio content by merging a person's facial features with another individual's. This process leverages advanced neural networks, notably GANs, which consist of two competing networks—the generator and the discriminator—to create hyper-realistic, yet entirely synthetic, media. While deepfakes have demonstrated potential applications in entertainment, education, and the arts, their widespread availability also presents considerable risks, including spreading misinformation, privacy violations, and

Corresponding author
Junkai Yi, yijk@bistu.edu.cn

a general erosion of trust in digital media. Consequently, researchers are intensively developing algorithms to detect deepfake videos, given the urgency of mitigating these risks (*Jiang et al., 2020*; *Li et al., 2020b*; *Rossler et al., 2019*; *Kong, Li & Wang, 2023*; *Kong et al., 2025*). In the early stages of deepfake detection research, many methods relied on spatial artifacts in images, such as unnatural details (*Liu et al., 2023*) and frequency-level checkerboard effects (*Marra et al., 2019*), which showed promising performance in single-frame detection tasks. However, these spatial artifact-based methods faced significant challenges when applied to deepfake videos, which consist of multiple frames, as they were less effective at capturing temporal artifacts. To overcome this limitation, recent studies (*Gu et al., 2021*; *Sabir et al., 2019*) have incorporated temporal cues, such as flickering (*Gu et al., 2022*; *Zheng et al., 2021*) and discontinuities (*Haliassos et al., 2021*), to improve the accuracy of deepfake video detection. Despite these advancements, studies (*Ojha, Li & Lee, 2023*) have demonstrated that detection methods relying on visual and temporal artifacts suffer from performance degradation when applied to newer high-quality generation techniques (*Dhariwal & Nichol, 2021*; *Ramesh et al., 2021*), as these advanced algorithms are designed to effectively suppress such artifacts. Nevertheless, most current deepfake detection methods (*Li, 2018*; *Qian et al., 2020*; *Rossler et al., 2019*; *Sabir et al., 2019*; *Yang, Li & Lyu, 2019*; *Zhao et al., 2021a*; *Masi et al., 2020*) perform well within the same dataset, but they often suffer from significant performance degradation when tested on different datasets. This degradation is primarily due to the distributional differences between the training and testing data. In real-world deepfake detection scenarios, which are characterized by high unpredictability and complexity, a reliable detector must possess strong generalization ability. However, each forgery method typically exhibits unique characteristics, and overfitting to a specific forgery type can severely hinder the model's ability to generalize to other types effectively. To address the challenges of data imbalance and the need for strong generalization in deepfake detection, this study first utilizes Focal Loss to assign higher weights to hard-to-classify samples, thereby alleviating the imbalance in data. This allows the model to focus on distinguishing between real and fake samples, especially when the distribution of real and fake samples is uneven. Next, ArcFace loss is introduced, which enhances the model by incorporating angular margins in the feature embedding space, strengthening the separation between classes and clarifying the boundaries between real and fake identities. This addition significantly improves the model's ability to distinguish real faces from manipulated ones across various datasets and manipulation techniques. Additionally, MixStyle simulates diverse visual styles through mixed feature statistics, offering a unique dimension for the model's generalization. This allows the model to adapt more effectively to variations in lighting, background, and appearance. By combining ArcFace loss, Focal Loss, and MixStyle, our approach not only achieves high accuracy but also demonstrates exceptional generalization across datasets and forgery types. This integrated method provides a comprehensive solution that strengthens both the model's discriminative power and adaptability, enabling accurate detection of deepfakes even against new, high-quality forgery techniques. The contributions of our work are summarized as follows:

- **We introduce a combined loss function**: By integrating Cross-Entropy Loss, Focal Loss, and ArcFace loss, we address both data imbalance and discriminative needs in face-centric tasks. Focal Loss helps the model focus on difficult-to-classify samples, while ArcFace loss enforces a clear margin between real and manipulated identities, resulting in more precise deepfake detection.
- **We enhance model generalization through MixStyle**: By incorporating MixStyle, our model gains robustness against variations in visual styles such as lighting, background, and facial expressions. MixStyle effectively simulates diverse conditions during training by mixing feature statistics between samples, which allows the model to better generalize to unseen conditions in real-world deepfake detection scenarios.
- **We conduct extensive experiments**: Our approach demonstrates state-of-the-art performance across various deepfake detection scenarios, including cross-dataset and cross-manipulation settings. Ablation studies further confirm the effectiveness of MixStyle and the loss function combination in improving classification accuracy.

# RELATED WORK

Over the past several years, significant progress has been made in the development of forgery creation techniques (*Gao et al., 2021*; *Jiang et al., 2021*; *Thies, Zollhöfer & Nießner, 2019*; *Kong et al., 2022*; *Luo et al., 2023*; *Kong, Wang & Li, 2022*), prompting increasing efforts to advance face forgery detection within the computer vision community. In this section, we review recent approaches for detecting generated images and videos. The previous methods primarily focus on enhancing the generalization ability of deepfake detection models, aiming to improve their performance across diverse datasets and forgery techniques.

## Deepfake generation

We broadly categorize face forgery algorithms into three groups: (1) Early attempts, such as landmark-based methods (*Bitouk et al., 2008*; *Wang et al., 2008*), which use face alignment to match the source face with a similar pose to the target face, followed by adjustments to color, illumination, and other factors to blend the source and target faces. (2) Subsequent studies (*Cao et al., 2013*; *Cheng et al., 2009*; *Dale et al., 2011*) introduced 3D face representations, including methods like Face2Face (*Thies et al., 2016*), face reproduction (*Suwajanakorn, Seitz & Kemelmacher-Shlizerman, 2017*), and expression manipulation (*Masi et al., 2016*; *Masi et al., 2019*), which enabled more realistic face manipulation. (3) In recent years, GAN-based face-switching methods have achieved even more realistic effects. These methods not only handle complex variations in illumination, texture, and angles but also generate high-quality images, enhancing the visual realism of the forgery. For instance, the DeepFakes approach customizes a model for each target person and trains it exclusively on that person's image data. On the other hand, FaceShifter (*Li et al., 2019*) can perform face swapping between any two faces without the need for prior training on the target person, thus improving the generalization and applicability of the model.

## Conventional deepfake detection

In deep forgery detection research, two primary types of detectors have emerged: image-level and video-level approaches. Image-level detectors focus on identifying fake images by detecting spatial artifacts within individual frames. For instance, Face X-ray (*Li et al., 2020a*) uses the boundaries between altered faces and their backgrounds to identify spatial inconsistencies, while Local relation learning (LRL) (*Chen et al., 2021*) combines RGB image analysis with frequency-aware cues to enhance detection capabilities. However, these image-based methods have a notable limitation as they fail to exploit the temporal cues inherent in deepfake videos. In contrast, video-level detectors leverage temporal information across multiple frames to identify manipulated content. For example, LipForensics (*Haliassos et al., 2021*) focuses on features from the mouth region, using a model pre-trained on a LipReading dataset to capture temporal dynamics. The fully temporal convolution network (FTCN) (*Zheng et al., 2021*) employs 3D CNNs with a spatial kernel size of 1 to directly extract temporal information. AltFreeze (*Wang et al., 2023b*), on the other hand, achieves robust generalization by independently training on both spatial and temporal features, enhancing its adaptability across different datasets and deepfake techniques.

## Deepfake detection generalization

With the rapid advancements in face generation and manipulation techniques, many earlier face forgery detection methods (*Rossler et al., 2019*; *Chai et al., 2020*; *Dong et al., 2023a*; *Luo et al., 2024*) struggle to handle previously unseen manipulations and datasets effectively. Moe-FFD (*Kong et al., 2024b*) introduces a mixture of experts model that improves face forgery detection efficiency and adaptability while reducing parameters. The task of deep forgery detection is deeply related to the generalization problem. The domain of image forgery detection has seen the development of innovative solutions from various approaches, including data augmentation (*Shiohara & Yamasaki, 2022*), frequency clues (*Gu et al., 2022*; *Liu et al., 2021*; *Qian et al., 2020*), ID information (*Dong et al., 2023b*), disentanglement learning (*Yan et al., 2023*), specifically designed networks (*Zhao et al., 2021a*) and 3D decomposition (*Zhu et al., 2021*). Video forgery detection efforts are focused on the temporal inconsistency (*Wang et al., 2023b*), eye blinking (*Li, Chang & Lyu, 2018*), and optical flow (*Amerini et al., 2019*).

# MATERIALS AND METHODS

## Code repository

To promote transparency and reproducibility in research, we have made the source code of the methods described in this paper publicly available for academic research and practical reference.

The code is available on GitHub at: https://github.com/13864084796/GFADE and can also be accessed online *via*: https://doi.org/10.5281/zenodo.15104245.

## MixStyle

Deepfake detection often faces challenges due to significant variations in visual conditions, such as differences in lighting, backgrounds, and textures across datasets. With limited

datasets, models are prone to overfitting, especially when they incorrectly generalize domain-specific information from a particular dataset as class-specific features. This leads to challenges when handling cross-domain datasets, as variations in lighting, backgrounds, textures, and even differences in capturing devices and storage mediums can introduce noise. These changes can cause the model to overfit specific dataset characteristics, limiting its generalizability to unseen data. To address this issue, our study employs MixStyle (*Zhou et al., 2021a*), a statistical mixing technique that enhances the model's robustness by introducing a broader range of visual conditions during training. Unlike forgery style mixing methods (*Kong et al., 2024a*), which operate at the forgery domain level, our MixStyle method introduces a novel approach by performing style mixing at the instance level through randomly pairing samples within the batch. This approach more precisely enhances style diversity, enabling the model not only to learn variations among different forgery styles but also to capture feature differences between real and forged samples. MixStyle is lightweight, model-agnostic, and can be seamlessly integrated into both convolutional neural network (CNN) and Transformer architectures. By disentangling style variations from core forgery features, MixStyle allows the model to focus on the intrinsic manipulation traces of deepfake content, reducing overfitting to dataset-specific styles and improving generalization to unseen forgery samples.

In the MixStyle method, we adopt a global shuffling strategy, where the entire batch is randomly reordered instead of following the original approach, which first divides the batch into multiple mini-batches and then performs shuffling and mixing within each group. Specifically, we randomly permute the batch and mix the style statistics using the newly assigned mean and standard deviation. This simplified approach reduces computational complexity while preserving the core objective of MixStyle—blending style-related feature statistics across different samples to enhance the model's generalization ability.

MixStyle operates within the intermediate layers of the neural network, adjusting the mean and standard deviation of feature maps to mix style-related feature statistics across samples. For a given batch with feature maps $f \in \mathbb{R}^{B \times C \times H \times W}$, where $B$ represents the batch size, $C$ is the channel count, and $H$ and $W$ are the height and width of the feature maps, the MixStyle process proceeds as follows:

**Compute instance statistics**: For each instance in the batch, the mean $\mu$ and standard deviation $\sigma$ of the feature maps are computed over the spatial dimensions $H$ and $W$. Specifically, for a feature map $f \in \mathbb{R}^{B \times C \times H \times W}$, where $B$ is the batch size, $C$ is the number of channels, and $H$ and $W$ represent the height and width of the feature maps respectively, the instance statistics for each feature map are calculated as:

$$\mu(f) = \frac{1}{H \times W} \sum_{h=1}^{H} \sum_{w=1}^{W} f_{:,:,h,w} \tag{1}$$

$$\sigma(f) = \sqrt{\frac{1}{H \times W} \sum_{h=1}^{H} \sum_{w=1}^{W} \left(f_{:,:,h,w} - \mu(f)\right)^2}. \tag{2}$$

**Mix statistics between instances**: After calculating the instance statistics, we randomly permute the entire batch before pairing instances for style mixing. Given two feature maps $f_i$ and $f_j$, the mixed statistics $\hat{\mu}$ and $\hat{\sigma}$ are computed as follows:

$$\hat{\mu} = \lambda \cdot \mu(f_i) + (1 - \lambda) \cdot \mu(f_j) \tag{3}$$

$$\hat{\sigma} = \lambda \cdot \sigma(f_i) + (1 - \lambda) \cdot \sigma(f_j) \tag{4}$$

where $\lambda$ is a randomly sampled mixing coefficient in the range $[0, 1]$.

**Apply mixed statistics**: After computing the mixed statistics, the next step is to apply these statistics to normalize the original feature maps. Specifically, for each feature map $f_i$ in the batch, the original feature maps are normalized using the mixed mean $\hat{\mu}$ and standard deviation $\hat{\sigma}$. The normalized feature map $\hat{f}$ is computed as follows:

$$\hat{f} = \hat{\sigma} \frac{f - \mu(f)}{\sigma(f)} + \hat{\mu}. \tag{5}$$

By incorporating mixed-style statistics from various samples, MixStyle helps the model simulate diverse visual environments during training. This approach encourages the model to learn representations that are less dependent on specific dataset styles, thereby improving its performance across different datasets and scenarios.

Additionally, high-quality forged samples are often visually similar to real samples, presenting a significant challenge in classification accuracy. MixStyle addresses this issue by reinforcing the model's sensitivity to subtle feature differences, which enhances its ability to detect nuanced variations between forged and real samples. This method prevents the model from overfitting to specific dataset biases, significantly improving its generalization capability and overall detection accuracy in deepfake detection tasks. It is important to note that MixStyle is applied only during training to enhance the model's generalization ability. During inference, feature statistics remain unchanged to ensure stable and consistent predictions.

## ArcFace loss

One of the biggest challenges in deepfake detection is that high-quality fake samples are visually very similar to real samples, making it difficult to distinguish them with the naked eye. In such cases, where "real and fake are extremely close", if we only use traditional binary classification loss functions (such as cross-entropy), the model is likely to focus solely on the goal of "classifying fake samples as fake", without paying attention to how far apart real and fake samples are in the feature space. As a result, the model may learn a "fuzzy" decision boundary, and when the quality of forgeries improves further, they can easily blend with real samples and be misclassified. ArcFace loss (*Deng et al., 2019*) is designed to better "push apart" the distance between real and fake samples. The core idea behind its approach is as follows: We first normalize the feature vectors extracted by the model and the center vectors of each class (real/fake), and then compute the cosine similarity. For the correct class (*e.g.*, a real sample should lie near the "real" center), an additional margin is

introduced in the angle of the cosine similarity. This effectively tells the model, "In order to classify a real sample as real, not only should the cosine similarity be high, but it must also go one step further to meet the 'acceptable' threshold". The final loss is computed based on this cosine similarity with the added margin, forcing the model to continuously increase the distance between different classes (real/fake) during training. Although deepfake detection is a binary classification task, the model's discriminative capability relies not only on global classification decisions but also on constructing well-defined class boundaries in the feature space. For tasks like deepfake detection, where the boundary between real and fake samples is extremely subtle and the forgery techniques are continuously evolving, "pushing a little further" can determine whether a fake sample is accurately identified or mistakenly classified as real. ArcFace loss provides additional discriminative power at the feature level, effectively reducing the interference caused by high-quality forgeries that "lie close" to real samples, thus significantly improving the accuracy and robustness of the detection model. Especially when faced with multiple forgery techniques or cross-dataset scenarios, the larger margin established by ArcFace loss helps the model better handle unknown, high-fidelity fake samples.

In conventional classification, Softmax loss is often used, where the loss function for a given sample $x_i$ with feature representation $f_i$ is defined as follows:

$$\mathcal{L}_{\text{softmax}} = -\log \frac{e^{s \cdot \cos(\theta_{y_i})}}{\sum_{j=1}^{n} e^{s \cdot \cos(\theta_j)}} \tag{6}$$

where:

- $s$ is a scaling factor for numerical stability,
- $\theta_j$ denotes the angle between the feature vector and the class center.

Typically, the direct application of Softmax loss cannot effectively enforce distinct class separation in the feature space. In deepfake detection, where subtle identity cues are essential, such a loss can be insufficient for achieving the necessary level of discrimination between real and manipulated samples.

To address this, an angular margin $m$ is introduced into the feature space, enhancing inter-class separation and reinforcing intra-class compactness. The improved loss function is given by:

$$\mathcal{L}_{\text{ArcFace}} = -\log \frac{e^{s \cdot \cos(\theta_{y_i} + m)}}{e^{s \cdot \cos(\theta_{y_i} + m)} + \sum_{j \neq y_i} e^{s \cdot \cos(\theta_j)}} \tag{7}$$

where $\cos(\theta_{y_i} + m)$ represents the calculated angular margin for the sample $x_i$ with respect to its true class $y_i$. By adding an angular margin $m$, ArcFace loss establishes a clearer decision boundary in the feature space, making real and fake identity representations more compact and distinct.

Introducing an angular margin is particularly effective in deepfake detection, where high-quality forgeries can closely mimic real visual features, making them difficult to distinguish using traditional methods. By optimizing the angular margin with ArcFace loss, the model can more effectively identify subtle differences in identity features. Specifically,

ArcFace loss enforces a separation between real and fake identities in feature space, providing clearer distinctions and reducing the risk of misclassification.

Moreover, ArcFace loss excels in generalization across diverse datasets. Deepfake generation techniques often introduce unique forgeries, each with distinct characteristics. A model without angular margin constraints may struggle to adapt to these variations, resulting in performance degradation across different forgery methods. The angular margin design, however, ensures stable class separation across a wide range of deepfake techniques, reducing the model's reliance on specific forgery characteristics. This enhances the model's generalization ability, improving cross-dataset and cross-technique performance. Therefore, ArcFace loss is a crucial tool for bolstering the robustness and discriminative power of deepfake detection models.

## Focal loss

In deepfake detection tasks, a significant challenge arises from the imbalance between hard-to-classify and easily classified samples. This imbalance can cause the model to be biased toward the majority class, reducing its ability to accurately detect samples. Hard-to-classify samples include real faces and high-quality forged samples that closely resemble real ones in appearance. Traditional classification methods, such as those using Cross-Entropy Loss, often struggle in this scenario as they do not account for class imbalance. The standard Cross-Entropy Loss function treats all samples equally, regardless of class distribution, which can lead to performance degradation when the majority class significantly outweighs the minority class. Since more realistic forged samples are harder to distinguish, Focal Loss assigns higher weights to these samples, guiding the model to focus on subtle distinguishing features. This helps mitigate the model's bias toward easily classified forged samples while enhancing its capability to detect complex deepfake cases.

The Cross-Entropy Loss is defined as:

$$\mathcal{L}_{\text{CE}} = -\sum_{i=1}^{N} y_i \log(p_i) \tag{8}$$

where $y_i$ is the true label for sample $i$, and $p_i$ represents the model's predicted probability for the true class. While Cross-Entropy Loss performs effectively under balanced conditions, it may fail to focus adequately on the minority class (real samples) in imbalanced datasets.

To mitigate this issue, Focal Loss (*Lin, 2017*) is employed by introducing a modulating factor $(1 - p_t)^\gamma$, where $p_t$ is the predicted probability for sample $t$. This factor adjusts the weight of each sample, encouraging the model to focus more on hard-to-classify samples. The Focal Loss function is defined as:

$$\mathcal{L}_{\text{Focal}} = -\sum_{i=1}^{N} \alpha_t (1 - p_t)^\gamma \log(p_t) \tag{9}$$

where:

- $\alpha_t$ is a balancing factor that controls the class weight, ensuring that minority classes receive sufficient attention in the loss calculation.

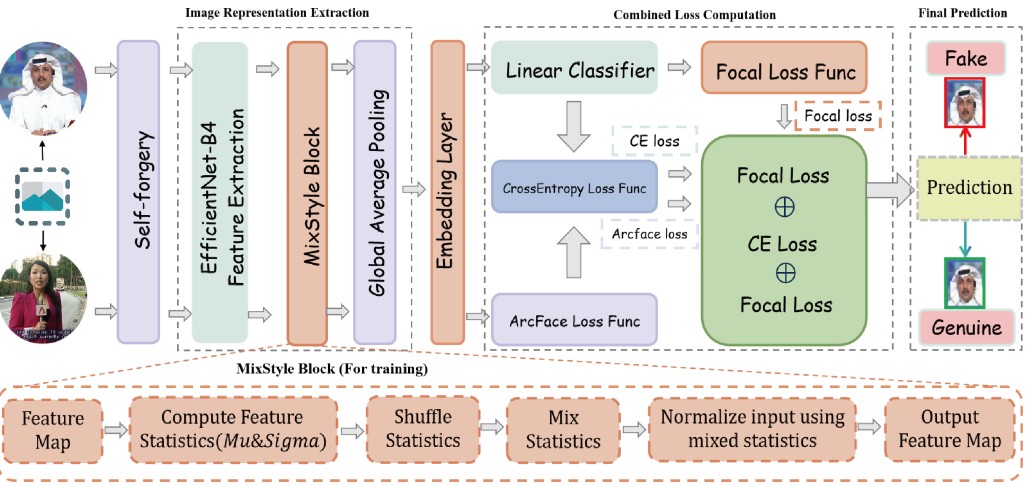

**Figure 1 Overall framework diagram of the generalized feature adaptation and discrimination enhancement (GFADE) model.**

- $\gamma$ is the focusing parameter, which increases the weight of hard samples (those that the model finds challenging to classify) by reducing the contribution of easily classified samples.

The term $(1 - p_t)^\gamma$ specifically diminishes the loss contribution from samples that the model classifies with high confidence (*i.e.*, samples with $p_t$ close to 1), thereby focusing more on misclassified or challenging samples. This effect directs the model's attention toward the minority class (real samples) and helps it capture subtle differences between real and forged instances.

Focal Loss proves particularly beneficial in deepfake detection where high-quality forged samples often visually resemble real samples, making them challenging to distinguish. By enhancing focus on hard-to-classify samples, Focal Loss allows the model to prioritize subtle distinctions within real samples, thereby improving its robustness and accuracy in identifying genuine instances within a highly imbalanced dataset.

## Overall framework

The proposed deepfake detection model framework starts by sampling frames from real videos, then applies self-forgery techniques (*Shiohara & Yamasaki, 2022*) to generate manipulated samples. During training, only real videos are utilized to construct both real and forged samples. The framework then performs multi-layer feature extraction and optimization, enhancing the model's ability to accurately detect and robustly classify deepfake content, as illustrated in Fig. 1.

First, the model randomly samples a set of key frames from the input video to ensure that any potential manipulation traces are captured. After each frame is sampled, the model applies self-augmentation to create synthetic manipulated frames. This self-forgery simulates various operations present in a deepfake generation, allowing the model to learn a broader range of manipulation patterns and features during training. The inclusion of

these augmented frames effectively increases the diversity of training samples, improving the model's generalizability to different manipulation types.

During the feature extraction stage, the model employs a pre-trained feature extraction network (*e.g.*, EfficientNet) to encode the self-augmented frames, generating high-dimensional feature representations. Subsequently, the model applies "the MixStyle" strategy to further improve generalization. MixStyle introduces diversity in the feature space by mixing feature statistics (*e.g.*, mean and standard deviation) , enabling the model to adapt to variations in lighting, background, and textures. This process enhances the model's robustness across different visual scenes, making it more stable in cross-domain and cross-dataset detection tasks.

In terms of loss function design, the model combines Cross-Entropy Loss, Focal Loss, and ArcFace loss to address class imbalance and distinguish between real and manipulated samples. Specifically:

- **Cross-entropy loss**: It is utilized to assess the difference between the model's predictions and the actual labels, offering a fundamental basis for optimization in classification tasks.
- **Focal loss**: In situations with class imbalance, Focal Loss assigns higher weights to hard-to-classify samples, reducing the influence of easily classified samples on the total loss, thereby improving the model's attention to minority manipulated samples.
- **ArcFace loss**: Introduces angular margins in the feature space to ensure a distinct separation between real and manipulated samples in the embedding space. The angular margin mechanism of ArcFace loss strengthens the model's discriminative capability between real and manipulated identities, particularly for fine-grained face forgery detection.

The combined loss function is defined as:

$$
\mathcal{L}_{total} = \mathcal{L}_{\text{CE}} + \mathcal{L}_{\text{Focal}} + \mathcal{L}_{\text{ArcFace}}
$$
$$
= -\sum_{i=1}^{N} \alpha_t (1-p_t)^{\gamma} \log(p_t) - \sum_{i=1}^{N} y_i \log(p_i) - \log \frac{e^{s \cdot \cos(\theta_{y_i}+m)}}{e^{s \cdot \cos(\theta_{y_i}+m)} + \sum_{j \neq y_i} e^{s \cdot \cos(\theta_j)}} \tag{10}
$$

where $\mathcal{L}_{\text{CE}}$ denotes Cross-Entropy Loss, $\mathcal{L}_{\text{Focal}}$ denotes Focal Loss, and $\mathcal{L}_{\text{ArcFace}}$ represents ArcFace loss.

In each training iteration, the model generates predictions for the self-augmented frames and computes Cross-Entropy Loss, Focal Loss, and ArcFace loss. These three losses are summed to form the total loss used for backpropagation. Meanwhile, MixStyle performs style mixing in the intermediate feature space, further enhancing the model's generalization across various visual styles.

## Experimental settings
### Computational environment
The experiments were conducted on a system equipped with one NVIDIA RTX 4090 GPU (24GB) and a CPU with 22 virtual cores, utilizing the AMD EPYC 7T83 64-core Processor. The software environment consisted of PyTorch version 1.11.0 running on Python 3.8 with Ubuntu 20.04, and Cuda version 11.3 was used to leverage the GPU's capabilities for efficient training and evaluation.

### Experimental datasets

We utilize the commonly referenced FaceForensics++ (FF++) (*Rossler et al., 2019*) benchmark for training, in line with established practices. This dataset comprises 1,000 original videos alongside 4,000 fabricated videos, generated through four distinct manipulation techniques: Deepfakes (DF), Face2Face (F2F) (*Thies et al., 2016*), FaceSwap (FS) (*Khodabakhsh et al., 2018*), and NeuralTextures (NT) (*Thies, Zollhöfer & Nießner, 2019*).The video resolution covers several levels (original quality, compressed quality C23 and C40) to evaluate the robustness of the method under different compression conditions. To evaluate the robustness of the proposed approach, particularly its ability to handle high-quality deepfakes, we test the trained model on five datasets that incorporate varying levels of deepfake quality. Celeb-DFv2 (CDF) (*Li et al., 2020b*) utilizes advanced deepfake generation techniques to create modified celebrity videos sourced from YouTube. This approach minimizes common visible artifacts, including fuzzy edges and color mismatches, enhancing the overall realism. The DeepFakeDetection dataset (DFD) comprises thousands of deepfake videos produced with the voluntary participation of actors. These videos were captured in controlled environments, ensuring diversity in scenes and lighting conditions. Additionally, the DeepFake Detection Challenge Preview (DFDCP) (*Dolhansky, 2019*) and the public test set (DFDC) (*Dolhansky et al., 2020*), released as part of the competition, include a variety of videos containing artifacts like compression, downsampling, and noise. FFIW-10K (FFIW) (*Zhou et al., 2021b*) focuses on scenarios involving multiple individuals; it is suitable for evaluating the performance of deepfake detection in multi-person interaction scenarios.

The datasets are available for deepfake detection research: FaceForensics++ (https://github.com/ondyari/FaceForensics), Celeb-DF-v2 dataset (https://github.com/yuezunli/celeb-deepfakeforensics), DFDC (https://dfdc.ai/login), DFDCP (https://dfdc.ai/login), and FFIW10K-v1 (https://github.com/tfzhou/FFIW).

### Evaluation indicators

The area under the curve (AUC) is a commonly used metric for evaluating the performance of binary classification models. It measures the area under the receiver operating characteristic (ROC) curve. The ROC curve plots the true positive rate (TPR) against the false positive rate (FPR) at various classification thresholds.

The TPR, also referred to as sensitivity (SEN), is defined as:

$$TPR = \frac{TP}{TP + FN} \tag{11}$$

where TP represents the true positives and FN represents the false negatives.

The FPR is defined as:

$$FPR = \frac{FP}{FP + TN} \tag{12}$$

where FP represents the false positives and TN represents the true negatives.

The AUC value summarizes the overall performance of the classifier, with a value ranging between 0.5 (indicating random classification) and 1 (perfect classification). A higher AUC signifies better discriminative power of the model.

While AUC provides a measure of a classifier's discriminative ability, additional metrics such as equal error rate (EER), and average precision (AP) further enhance the evaluation.

The EER is commonly used in biometric verification systems and represents the point at which the false acceptance rate (FAR) equals the false rejection rate (FRR). It is defined as:

$$\text{FAR} = \frac{\text{FP}}{\text{FP} + \text{TN}}, \quad \text{FRR} = \frac{\text{FN}}{\text{TP} + \text{FN}} \tag{13}$$

EER is obtained by identifying the threshold where FAR and FRR intersect, indicating a balance between false acceptances and false rejections.

Range: [0,1]. Ideal Condition: A value close to **0** indicates better classification performance, minimizing errors.

Average precision (AP) evaluates a classifier's precision–recall tradeoff and is computed as the area under the precision-recall (PR) curve:

$$\text{AP} = \sum_n (\text{Recall}_n - \text{Recall}_{n-1}) \cdot \text{Precision}_n \tag{14}$$

where:

$$\text{Precision} = \frac{\text{TP}}{\text{TP} + \text{FP}}, \quad \text{Recall} = \frac{\text{TP}}{\text{TP} + \text{FN}}. \tag{15}$$

AP quantifies how well the model maintains high precision across varying recall levels.

Range: [0,1]. Ideal Condition: A value close to **1** indicates that the classifier maintains high precision at all recall levels.

By incorporating these additional metrics alongside AUC, a more comprehensive evaluation of the classifier's performance can be achieved, ensuring robust assessment across various classification scenarios.

### *Experimental configuration*

In our experiments, the model was trained for 100 epochs with a batch size of 32. The input images were resized to 380 × 380 pixels to balance computational complexity and visual detail. The initial learning rate was set to 0.001, and a learning rate scheduler (LinearDecayLR) was used to gradually decay the learning rate after 75 epochs, ensuring rapid convergence in the early stages while fine-tuning in later epochs. We employed the Sharpness-Aware Minimization (SAM) optimizer combined with SGD (Stochastic Gradient Descent), configured with a learning rate of 0.001, momentum of 0.9, and weight decay to enhance the model's generalization ability. The primary loss function was Cross-Entropy Loss, complemented by Focal Loss to address class imbalance and ArcFace loss to enhance the model's discriminative power by introducing angular margins. Data augmentation techniques, including random horizontal flipping and random rotation (up to 15 degrees), were applied to increase the diversity of the training data and reduce overfitting risks.

### Model workflow
### *Preprocessing*

In this study, we employ Dlib and RetinaFace for face detection and preprocessing, with Dlib being used during the training phase and RetinaFace during the inference phase.

During training, the Dlib 81 facial landmark predictor is utilized to locate facial features in each video frame accurately, and facial bounding boxes are generated to calculate the face width and height. Based on these bounding boxes, a random margin of 4% to 20% is applied to crop the face region, enhancing the model's robustness and generalization ability. The cropped facial images and corresponding landmark data are then stored for subsequent analysis. For inference, to improve computational efficiency, we rely solely on the RetinaFace model, which uses a ResNet-50 architecture to detect facial bounding boxes and five key facial landmarks. A fixed margin of 12.5% is applied to crop the face region during inference, without additional landmark detection. Video frames are uniformly sampled according to predefined parameters and converted to the appropriate color space to match the model input. Additionally, the detected faces are sorted by size, prioritizing the largest face in each frame for processing. All processed images and data are saved in a structured file hierarchy to ensure efficient data management during both training and inference. This approach ensures diversity during training while significantly improving computational efficiency during inference.

### Training

During this training process, we utilized EfficientNet-b4 as the classifier, which had been pre-trained on the ImageNet dataset. The model was trained using the SAM optimizer for a total of 100 epochs, with a batch size of 32 and an initial learning rate of 0.001. To further enhance the model's convergence, a learning rate scheduler (LinearDecayLR) was applied, which kept the learning rate constant for the first 75% of the training epochs, followed by a gradual decay during the remaining epochs. Regarding video processing, 32 frames were randomly sampled from each video for training purposes. In cases where multiple faces were detected in a single frame, the face with the largest bounding box was selected for feature extraction. To improve data loading efficiency, multi-threading (four worker threads) was utilized to generate batches in parallel, and pin_memory was enabled to expedite data transfer from the CPU to the GPU.

### Inference

During the inference phase, 32 frames are randomly sampled from each video for processing. For each frame, a pre-trained ResNet-50 face detector is employed to detect faces. If multiple faces are detected in a single frame, the classifier generates predictions for each detected face, and the prediction with the highest fakeness confidence is selected as the final prediction for that frame. To obtain the final prediction score for the video, the average of the predictions across all frames is calculated. To ensure model robustness, special handling is implemented for potential anomalies during inference. If no faces are detected in any frames of a video, the confidence score for that video is set to 0.5, ensuring that all videos in the test set are included in the evaluation. This approach ensures fairness in the ROC-AUC calculation.

**Table 1** **In-dataset and cross-dataset evaluation in terms of AUC (%), EER (%) and AP (%) on multiple deepfake datasets.** Bold and underlined highlight the best and the second-best performance, respectively.

| Method | FF++ | | | CDF2 | | | DFD | | | DFDC | | |
|---|---|---|---|---|---|---|---|---|---|---|---|---|
| | AUC | EER | AP | AUC | EER | AP | AUC | EER | AP | AUC | EER | AP |
| Xception (*Rossler et al., 2019*) | 93 | 3.77 | – | 65.27 | 38.77 | – | 87.86 | 21.04 | – | 69.90 | 35.41 | – |
| Self-blended image (SBI) (*Shiohara & Yamasaki, 2022*) | 99.64 | – | – | 93.18 | – | 85.16 | 97.56 | – | 89.49 | 76.15 | – | 93.24 |
| AltFreezing (*Wang et al., 2023b*) | 98.6 | – | – | 89.50 | – | – | 98.50 | – | – | – | – | – |
| EN-b4 (*Tan & Le, 2019*) | 99.22 | 3.36 | – | 68.52 | 35.61 | – | 87.37 | 21.99 | – | 70.12 | 35.54 | – |
| Fully temporal convolution network (FTCN) (*Zheng et al., 2021*) | – | – | – | 86.90 | – | – | 94.40 | – | – | 71.00 | – | – |
| Face X-ray (*Li et al., 2020a*) | 87.40 | – | – | 74.20 | – | – | 85.60 | – | – | 70.00 | – | – |
| RECCE (*Cao et al., 2022*) | 97.06 | – | – | 70.93 | – | 70.35 | 98.26 | – | 79.42 | – | – | – |
| Multi-attentional (*Zhao et al., 2021a*) | 99.27 | 3.35 | – | 68.26 | 32.83 | 75.25 | 92.95 | 21.73 | 96.51 | 67.34 | 38.31 | – |
| PCL+I2G (*Zhao et al., 2021b*) | 99.11 | – | – | 90.03 | – | – | 99.07 | – | – | 74.27 | – | – |
| SFGD (*Wang et al., 2023a*) | 99.53 | – | – | 75.83 | – | – | 88.00 | – | – | 73.63 | – | – |
| Critical Forgery Mining (CFM) (*Luo et al., 2023*) | 99.62 | – | – | 89.65 | 17.65 | – | 70.59 | 35.02 | – | 80.22 | 27.48 | – |
| CADDM (*Dong et al., 2023a*) | **99.79** | – | – | 93.88 | – | 91.12 | 99.03 | – | 99.59 | – | – | – |
| **GFADE (Ours)** | 99.68 | 2.14 | 99.93 | 94.85 | 12.92 | 97.2 | 99.12 | 13.41 | 99.61 | 80.81 | 26.68 | 95.97 |

## RESULTS AND DISCUSSION

In this section, we analyze the performance of our proposed method against state-of-the-art techniques, assessing both qualitative and quantitative results. Finally, we perform an ablation study to evaluate the impact of various components within our model.

### Cross-dataset evaluation

To demonstrate the robustness and generalizability of our approach, we perform a cross-dataset evaluation. In this process, models are trained on the FaceForensics++ (FF++) dataset, a widely acknowledged benchmark known for its extensive collection of manipulated videos, which is commonly used for deepfake detection. However, achieving high performance on the training dataset does not guarantee that the model will perform effectively on unseen data from different sources or generated by alternative deepfake techniques. To address this, we evaluate the trained models on external datasets, specifically CDF2, DFD, and DFDC, which include a diverse range of deepfake generation methods, manipulation styles, and varying video quality levels. The models' performance is measured using three critical metrics: AUC, EER, and AP, as summarized in Table 1.

The generalized feature adaptation and discrimination enhancement (GFADE) model demonstrates excellent performance across multiple deepfake datasets (FF++, CDF2, DFD, DFDC), showcasing its strong capabilities. In terms of AUC, GFADE achieves outstanding results on all datasets, particularly reaching 94.85% on the CDF dataset, highlighting its remarkable ability to distinguish between real and fake images. In terms of EER (Equal Error Rate), GFADE also performs excellently, with values of 2.14% on FF++ and 12.92% on CDF2, indicating a low false-positive rate and high classification accuracy. For AP,

**Table 2** **Generalization performance on low-quality videos.** Bold indicates the best performance.

| Method | Raw | C23 | C40 |
|---|---|---|---|
| FTCN (*Zheng et al., 2021*) | 99.5 | 99.0 | 70.2 |
| Style Latent (*Choi et al., 2024*) | 99.6 | **99.1** | 71.8 |
| **GFADE (Ours)** | **99.68** | 92.57 | **76.29** |

GFADE reaches a high value of 99.93% on the FF++ dataset, demonstrating its powerful ability to accurately identify forged images. GFADE also exhibits excellent generalization across different datasets, further validating its wide applicability and stability.

Furthermore, the performance of the GFADE model was evaluated in terms of AUC on the DFDCP (85.39%) and FFIW (85.55%) datasets. These results demonstrate a significant improvement compared to FTCN, which achieved AUC scores of 74.0% on DFDCP and 74.47% on FFIW, underscoring the superior effectiveness of GFADE on these datasets.

## Generalization performance on low-quality videos

To evaluate the robustness of our proposed GFADE model, we conducted experiments to test its generalization capabilities on low-quality video scenarios. Table 2 presents the results of our model compared to baseline methods, specifically focusing on video compression levels represented by Raw, C23, and C40 versions of the dataset.

The comparison includes three methods: FTCN, Style Latent, and our proposed GFADE. The 'Raw' column represents the original high-quality video, while 'C23' and 'C40' denote videos compressed at different levels, with 'C40' being the most compressed and hence having the lowest quality.

The experimental results demonstrate the following:

- **Raw videos**: Our GFADE model achieves the highest AUC score of 99.68%, outperforming both FTCN (99.5%) and Style Latent (99.6%). This indicates the effectiveness of our approach in detecting forgeries in high-quality videos.
- **C23 compression level**: While Style Latent slightly outperforms our model at this compression level, achieving an AUC of 99.1% compared to GFADE's 92.57%, the results highlight the resilience of Style Latent under moderate compression. However, GFADE still demonstrates satisfactory performance, confirming its capability under such conditions.
- **C40 compression level**: The GFADE model significantly outperforms both FTCN and Style Latent, achieving an AUC of 76.29%. This suggests that GFADE is particularly robust under severe video compression, where details are lost, and detection becomes more challenging. The higher AUC score at this compression level reflects the effectiveness of our integration of MixStyle and advanced loss functions, which improve generalization to more challenging visual scenarios.

Overall, the results from Table 2 clearly illustrate the superior performance of our GFADE model, especially in challenging low-quality video conditions (C40), demonstrating its robustness in real-world scenarios where videos are often subject to significant compression and quality loss.

**Table 3  Cross-manipulation evaluation on FF++.** Bold values represent the best results.

| Method | Test Set AUC (%) | | | | |
|---|---|---|---|---|---|
| | DF | F2F | FS | NT | FF++ |
| Face X-ray (*Li et al., 2020a*) | 99.17 | 98.57 | 98.21 | 98.13 | 98.52 |
| PCL + I2G (*Zhao et al., 2021b*) | 99.99 | 98.97 | 99.86 | 97.63 | 99.11 |
| EFNB4 + SBIs (*Shiohara & Yamasaki, 2022*) | 99.99 | 99.88 | **99.91** | 98.79 | 99.64 |
| Style Latent (*Choi et al., 2024*) | 99.7 | 98.6 | 98.8 | 96.4 | 99.1 |
| **GFADE (Ours)** | **99.99** | **99.94** | 99.83 | **98.96** | **99.68** |

## Cross-manipulation evaluation

GFADE is trained on real samples from the FaceForensics++ (FF++) dataset as well as forged samples generated using self-forgery techniques (*Shiohara & Yamasaki, 2022*). To evaluate the model's generalization to other different forgery operations, we assess its performance on the four manipulation techniques provided in the FaceForensics++(FF++) dataset, including Deepfakes (DF), Face2Face (F2F), FaceSwap (FS), and NeuralTextures (NT). These diverse manipulation methods represent a wide range of deepfake generation techniques, each posing unique challenges for detection. By evaluating our model on these different manipulations, we aim to demonstrate its robustness and generalizability in detecting forged content across various manipulation types. This cross-evaluation ensures that the model is not overfitted to a specific manipulation technique and can effectively generalize to detect forgeries generated through different methods.

The results in Table 3 highlight the competitive performance of various models on cross-manipulation evaluation using the FaceForensics++ dataset. Among the listed methods, our proposed model, GFADE, achieves the best performance across most manipulation techniques, as indicated by the bold values. Specifically, GFADE attains the highest AUC scores for Deepfakes (99.99%), Face2Face (99.94%), and NeuralTextures (98.96%), demonstrating its robustness and superior generalizability. Notably, EFNB4 + SBIs achieves the best performance for FaceSwap (99.91%), marginally surpassing GFADE (99.83%). However, GFADE consistently outperforms other methods in overall performance on the FF++ dataset, achieving an AUC of 99.68%. This suggests that while some models may excel in specific manipulation techniques, GFADE provides a more balanced and robust detection capability across diverse manipulation methods.

The superior performance of GFADE can be attributed to its advanced feature extraction and generalization strategies, enabling it to effectively detect subtle artifacts across different deepfake generation techniques. Compared to existing methods such as Face X-ray and PCL + I2G, which also achieve high AUC scores, GFADE demonstrates a noticeable improvement, particularly for challenging manipulations like NeuralTextures. This indicates that GFADE is less prone to overfitting and more adaptable to various forgery patterns, which is critical for real-world applications in deepfake detection.

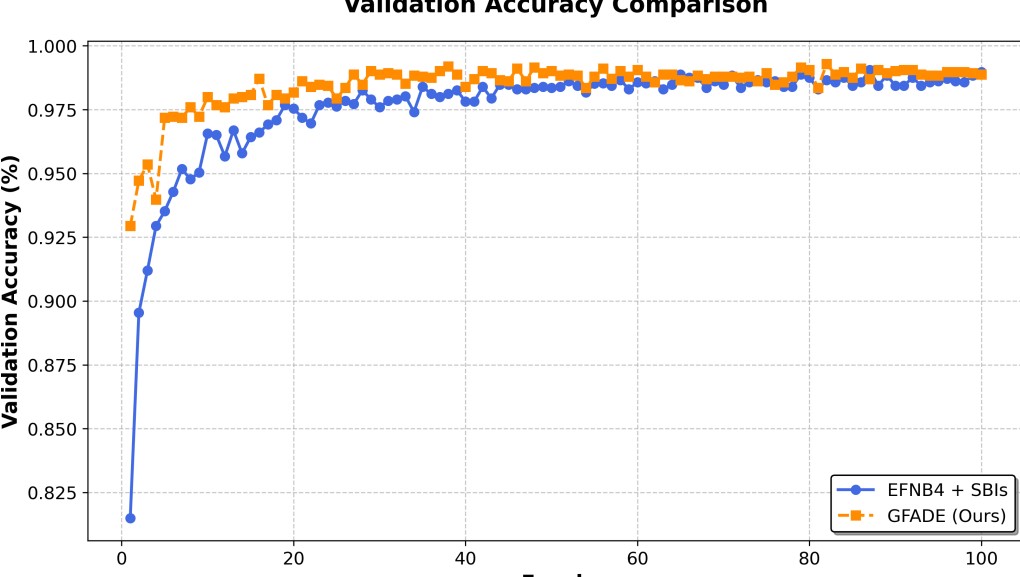

**Figure 2** Validation accuracy comparison between two models.

## Further analysis

To comprehensively evaluate our model's performance, we analyze the AUC curves and loss curves in comparison to baseline models, along with the impact of replacing different network architectures on the model's performance.

### AUC curve comparison with baseline models

In this section, we compare the AUC curves of our model with the baseline models across the training process. The AUC curves provide insight into the classification performance on the validation set, demonstrating the model's generalization ability. Our model consistently shows higher AUC values compared to the baseline, indicating superior performance in distinguishing between fake and real samples. This improvement in the AUC curve suggests that our model achieves better classification accuracy and maintains robust generalization across validation epochs, as illustrated in Fig. 2.

The accuracy of GFADE rises rapidly in the initial phase, significantly faster than that of self-blended image (SBI). This suggests that GFADE has better convergence properties and achieves near-peak performance in fewer training iterations. In the middle and late stages of training (after about 40 epochs), the validation accuracy curve of GFADE fluctuates less and shows more stability. This implies that GFADE is more robust to perturbations in training, as shown in Fig. 2.

### Loss curve comparison with baseline models

We also analyze the loss curves of our model in comparison to the baseline models throughout training (see Fig. 3). The loss curves illustrate convergence speed and stability, where our model demonstrates a faster decrease in loss during the early stages, indicating

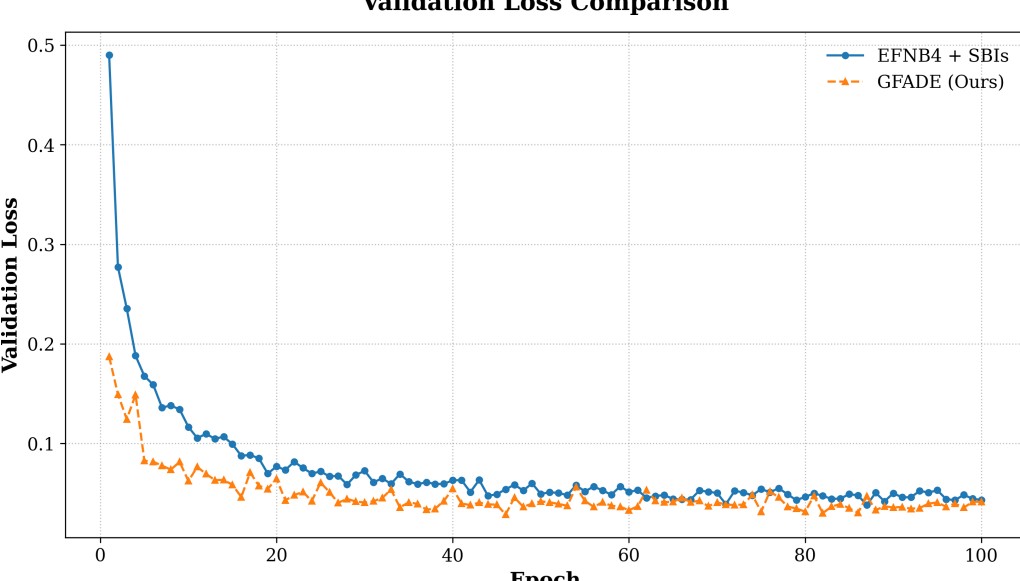

**Figure 3  Validation loss comparison between two models.**

higher optimization efficiency. This rapid convergence suggests that our model is able to learn the distinguishing features between real and fake samples more effectively, reaching a stable and lower final loss compared to the baseline. Overall, the comparison of loss curves further emphasizes the efficiency and effectiveness of our approach in minimizing training loss while achieving higher detection accuracy.

### Network architecture comparison

In addition to our primary experiments, we conducted a comparative analysis of various network architectures, including ResNet-50, ResNet-152, Xception, EfficientNet-b1, and EfficientNet-b4. The results, as illustrated in Fig. 4, highlight how our model performs across these architectures.The validation accuracy trends reveal clear distinctions in the convergence behavior and overall performance of the evaluated architectures. The EfficientNet series, particularly EfficientNet-b4, demonstrates a steady and rapid rise in accuracy during the early epochs, surpassing other models in achieving near-optimal accuracy. While ResNet-50 and ResNet-152 also achieve competitive results, their convergence rates are slower compared to EfficientNet-b1 and b4, indicating a potential trade-off between model complexity and optimization speed.

By evaluating our model on these different setups, we aim to determine the optimal balance between detection accuracy and computational overhead. As shown in Table 4, the experimental results highlight that our model exhibits excellent performance across all architectures, validating its strong generalization ability across different setups. This comprehensive analysis provides valuable insights into the trade-offs between computational cost and accuracy, helping us identify the most effective architecture for real-world applications. Among the evaluated architectures, EfficientNet-b4 consistently

**Figure 4** Validation accuracy comparison between different models.

**Table 4** Performance comparison of different network architectures across multiple test sets. Bold indicates the best performance.

| Architecture | Test Set AUC (%) | | | | |
|---|---|---|---|---|---|
| | FF++ | CDF | DFDCP | FFIW | Avg |
| ResNet-50 | 99.16 | 86.90 | 81.83 | 84.85 | 88.19 |
| ResNet-152 | 99.36 | 89.30 | 80.91 | 80.08 | 87.41 |
| Xception | 99.68 | 88.03 | 76.41 | 80.66 | 86.20 |
| EfficientNet-b1 | 99.59 | 90.27 | 82.11 | 82.13 | 88.53 |
| EfficientNet-b4 | **99.68** | **94.85** | **85.39** | **85.55** | **93.87** |

achieves the highest performance, with the best AUC scores across all test sets (FF++: 99.68%, CDF: 94.85%, DFDCP: 85.39%, FFIW: 85.55%) and an overall average AUC of 93.87%. This demonstrates its superior capability in balancing detection accuracy across diverse datasets.

## Ablation study

To comprehensively assess the contribution of each component in our framework, we divided the ablation study into two main parts: the impact of different loss function combinations and the effect of MixStyle.

### Effect of loss functions

In this section, we explore the impact of various loss function combinations on deepfake detection performance. We evaluated the following configurations: $\mathcal{L}_{CE}$: Serves as the

**Table 5 Ablation study results for different loss function configurations on various datasets.** Bold values represent the best results for each dataset.

| Loss configuration | Test Set AUC (%) | | | | Average AUC (%) |
|---|---|---|---|---|---|
| | FF++ | CDF | DFDC | FFIW | Avg |
| $\mathcal{L}_{CE}(Baseline)$ | 99.65 | 93.19 | 72.46 | 83.51 | 87.2 |
| $\mathcal{L}_{CE} + \mathcal{L}_{ArcFace}$ | 99.66 | 94.73 | 79.04 | 84.33 | 89.44 |
| $\lambda_1\mathcal{L}_{CE} + \lambda_2\mathcal{L}_{ArcFace} + \lambda_3\mathcal{L}_{Focal}$ | **99.66** | **94.81** | **79.45** | **84.84** | **89.69** |

baseline, providing a standard classification loss function without additional discriminative margins or focus on challenging samples. $\mathcal{L}_{CE} + \mathcal{L}_{ArcFace}$: Combines Cross-Entropy Loss with ArcFace loss, adding angular margins to improve the model's discriminative power by enforcing clearer boundaries between real and fake samples. $\mathcal{L}_{CE} + \mathcal{L}_{ArcFace} + \mathcal{L}_{Focal}$: This full configuration combines Cross-Entropy Loss, ArcFace loss, and Focal Loss to address both class imbalance and discriminative requirements, allowing the model to better handle hard-to-classify samples.

The results in Table 5 demonstrate the incremental improvements achieved by combining loss functions. While $\mathcal{L}_{CE}$ alone provides a strong baseline, its performance on challenging datasets like DFDC (72.46%) and FFIW (83.51%) is limited, highlighting its inability to handle complex scenarios. Adding $\mathcal{L}_{ArcFace}$ improves AUC scores across all datasets, particularly on DFDC (79.04%), by enhancing feature separability through angular margins. The inclusion of $\mathcal{L}_{Focal}$ further addresses class imbalance, leading to the highest average AUC (89.69%), with notable gains on DFDC and FFIW. This demonstrates that the combination of $\mathcal{L}_{CE}$, $\mathcal{L}_{ArcFace}$, and $\mathcal{L}_{Focal}$ effectively balances discrimination and focused learning, ensuring robust detection performance across diverse datasets.

To further analyze the impact of different loss components on model performance, we conduct an ablation study on the weighting of each loss term. While our approach integrates Cross-Entropy Loss, ArcFace loss, and Focal Loss to enhance classification and generalization capabilities, the relative contributions of these loss functions may vary depending on their assigned weights. A comprehensive investigation into the effect of different weighting strategies is crucial to optimizing the balance between classification accuracy, feature discrimination, and robustness against imbalanced data. Therefore, we systematically evaluate various weight configurations to determine their influence on deepfake detection performance.

Based on the experimental results in Table 6, the best model performance is achieved with the weight configuration of $\lambda_1 = 1$, $\lambda_2 = 1$, $\lambda_3 = 0.1$, resulting in the highest average AUC (89.69%). This configuration performs well across the FF++, CDF, and FFIW datasets, particularly improving detection performance on challenging datasets like DFDC and FFIW. The higher weight for ArcFace loss ($\lambda_1 = 1$) effectively enhances feature discriminability, especially improving accuracy on challenging datasets like DFDC and FFIW. The appropriate weight for Focal Loss ($\lambda_3 = 0.1$) helps improve the model's ability to classify difficult samples.

**Table 6 Ablation study on the impact of loss weighting (AUC%).** Bold indicates the best performance.

| | | | Average AUC (%) | | | | |
|---|---|---|---|---|---|---|---|
| $\lambda_1$ | $\lambda_2$ | $\lambda_3$ | FF++ | CDF | DFDC | FFIW | Avg |
| 0.1 | 0.1 | 0.1 | 99.11 | 93.89 | 78.99 | 78.95 | 87.74 |
| 1 | 1 | 1 | 99.37 | 94.45 | 79.01 | 84.12 | 89.24 |
| 10 | 10 | 10 | 99.20 | 93.32 | 78.23 | 84.23 | 88.75 |
| 0.1 | 1 | 1 | 99.50 | 94.61 | **79.47** | 84.69 | 89.57 |
| 1 | 0.1 | 1 | 99.45 | 94.59 | 79.17 | 84.46 | 89.42 |
| 1 | 1 | 0.1 | **99.66** | **94.81** | 79.45 | **84.84** | **89.69** |

**Table 7 Ablation study results for different configurations with MixStyle on various datasets.** Bold values represent the best results for each dataset.

| Configuration with MixStyle | Test Set AUC (%) | | | | Average |
|---|---|---|---|---|---|
| | FF++ | CDF | DFDC | FFIW | Avg |
| Baseline(SBI) | 99.65 | 93.19 | 72.46 | 83.51 | 87.2 |
| MixStyle | 99.62 | 94.02 | 79.03 | 84.26 | 89.23 |
| $\mathcal{L}_{CE} + \mathcal{L}_{ArcFace} +$ MixStyle | 99.67 | 94.82 | 80.73 | 85.55 | 90.20 |
| $\mathcal{L}_{CE} + \mathcal{L}_{ArcFace} + \mathcal{L}_{Focal} +$ MixStyle | **99.68** | **94.85** | **80.81** | **85.55** | **90.22** |

### Impact of MixStyle

To assess the effect of MixStyle on model performance and generalization, we experimented with the following configurations: MixStyle: Evaluates the standalone impact of MixStyle on visual generalization, enhancing adaptability to diverse conditions without additional discriminative losses. $\mathcal{L}_{CE} + \mathcal{L}_{ArcFace} +$ MixStyle: This configuration combines MixStyle with Cross-Entropy and ArcFace loss, enhancing both generalization and discriminative capability. $\mathcal{L}_{CE} + \mathcal{L}_{ArcFace} + \mathcal{L}_{Focal} +$ MixStyle: This full combination integrates MixStyle, Cross-Entropy Loss, ArcFace loss, and Focal Loss, maximizing generalization and classification accuracy.

The results, summarized in Table 7, indicate that the full combination with MixStyle and both loss functions yields the best performance, underscoring MixStyle's role in enhancing cross-dataset generalization when paired with strong discriminative and class-balancing loss functions.

The results demonstrate that each component contributes uniquely to the model's performance. The full configuration of $\mathcal{L}_{CE}$, $\mathcal{L}_{ArcFace}$, $\mathcal{L}_{Focal}$, and MixStyle achieves the highest results, highlighting the combined benefits of improved discrimination, visual generalization, and focus on hard samples.

### Visualization analysis

To intuitively understand the model's performance and decision-making process, we employ two visualization techniques: t-SNE feature space analysis and saliency map visualization.

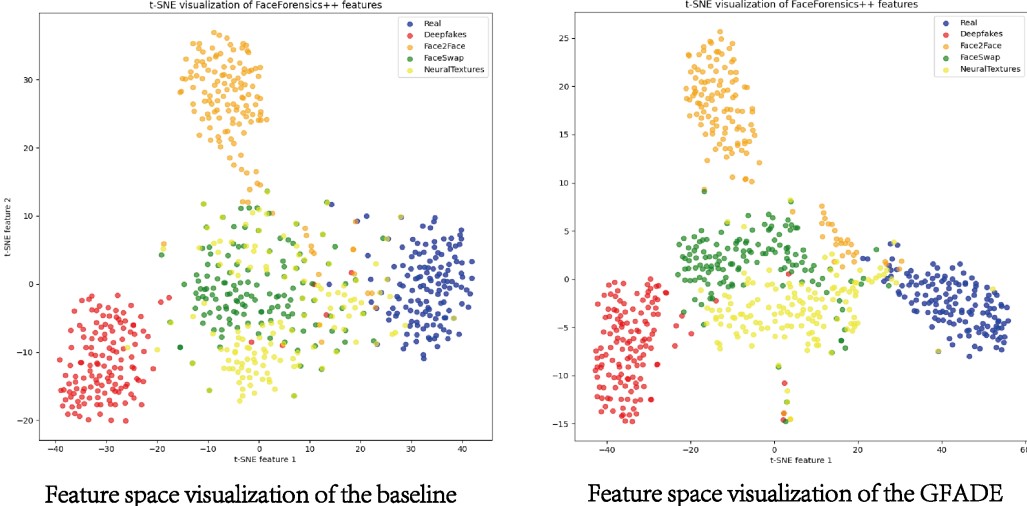

**Figure 5** **Feature space visualization of GFADE.** (Blue corresponds to Real, Red represents Deepfakes, Orange stands for Face2Face, Green indicates FaceSwap and Yellow signifies NeuralTextures.).

### *T-SNE feature space*

This t-SNE feature space analysis allows us to observe how real and fake video samples are clustered. In this study, we present a comparative analysis of t-SNE visualizations of features extracted from FaceForensics++ datasets using GFADE and the baseline. The GFADE model demonstrates a significant improvement in feature separation, as evidenced by the distinct clustering patterns observed in the t-SNE plot shown in Fig. 5. In the GFADE t-SNE visualization, data points from different classes form clearly separated clusters with minimal overlap, indicating a strong inter-class separation. This contrasts with the baseline's t-SNE plot, where there is substantial overlap between classes, resulting in blurred boundaries and inter-class ambiguity.

Additionally, the intra-class compactness of GFADE is evident, particularly within classes like Deepfakes (red), FaceSwap (green), and NeuralTextures (yellow), where points form tight, coherent clusters. As depicted in Fig. 5, this compactness suggests that GFADE extracts more consistent features within each class, reducing intra-class variability. Moreover, GFADE minimizes the presence of ambiguous or transition points between classes, which are prominent in the baseline's visualization. This reduction in ambiguous points likely enhances GFADE's reliability and robustness in real-world classification tasks, as it indicates more defined feature boundaries between real and manipulated content.

Overall, the improved feature separation and intra-class cohesion in GFADE's t-SNE plot suggest a more effective and discriminative feature extraction process, contributing to enhanced classification performance and accuracy in distinguishing between real and manipulated videos.

### *Saliency map*

To better interpret the model's decision-making process, we applied saliency map techniques to visualize the regions of the input frames that significantly influence the

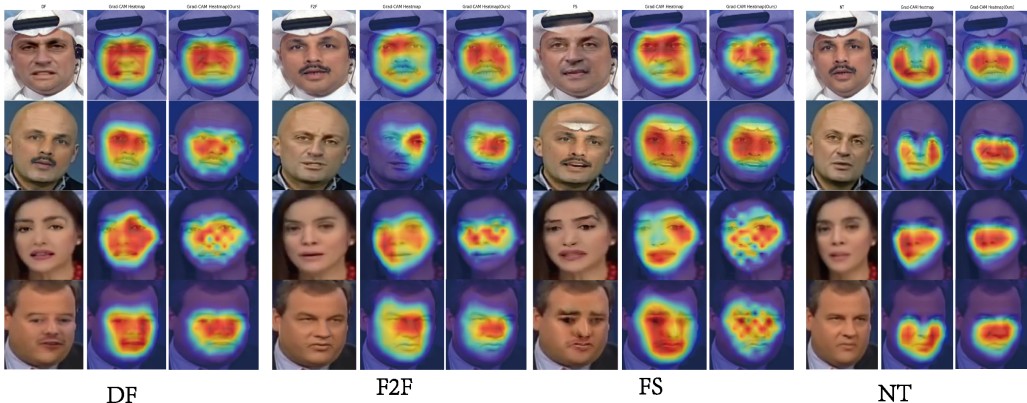

DF          F2F          FS          NT

**Figure 6** Saliency map visualization on DF, FS, F2F, and NT datasets highlights the comparative performance of the SBI model (second column) and our model (third column).

model's predictions. By highlighting pixel-level importance, saliency maps provide insights into the areas where the model focuses when detecting deepfake content. As illustrated in Fig. 6, our analysis shows that the model consistently attends to critical facial regions, such as the eyes, mouth, and jawline, where manipulation artifacts, such as unnatural textures or subtle misalignments, tend to appear. These visualizations not only validate the model's attention to key areas but also emphasize the importance of local facial details in identifying deepfake forgeries.

Furthermore, GFADE exhibits greater consistency across diverse samples, consistently capturing salient facial features, as shown in Fig. 6, which enhances the model's reliability and robustness in varied image contexts. Additionally, GFADE's heatmaps contain less noise and show greater concentration on key regions, resulting in stronger alignment with actual facial features. This alignment offers a more intuitive and reliable visual interpretation for end-users. Collectively, these findings suggest that GFADE outperforms the baseline model in its ability to accurately localize essential features, mitigate background noise, and enhance interpretability—advantages that significantly contribute to its discriminative power and practical applicability.

## CONCLUSION

In this article, we propose a robust and comprehensive deepfake detection framework that effectively integrates multiple loss functions with the MixStyle technique to address the challenges in deepfake detection. By leveraging ArcFace loss and Focal Loss, the proposed model significantly enhances its ability to distinguish between real and fake samples, even in the presence of high-quality forgeries and highly imbalanced datasets. The incorporation of MixStyle further improves the model's adaptability by introducing diverse visual styles during training, enabling superior generalization across unseen datasets and manipulation techniques.

Extensive experiments across multiple benchmark datasets demonstrate that our method achieves state-of-the-art performance, maintaining stable detection accuracy even under

cross-dataset and cross-manipulation scenarios. This highlights the effectiveness of the combined loss functions and MixStyle strategy in improving both discriminative power and generalizability.

### Future works

While the proposed framework shows promising results, several directions remain for future improvement. First, incorporating advanced style perturbations and self-supervised learning techniques could further enhance the model's resilience to emerging and more sophisticated forgery methods. Second, optimizing the model's computational efficiency for real-time deepfake detection in streaming video applications remains a crucial avenue for practical deployment. Lastly, expanding the evaluation to larger, more diverse datasets and real-world scenarios will help ensure the model's robustness and adaptability in broader contexts.

In summary, our study provides a strong foundation for improving deepfake detection, balancing accuracy, robustness, and generalization, with potential extensions into real-world, real-time applications.

## ACKNOWLEDGEMENTS

We would like to express our heartfelt gratitude to the authors of the FaceForensics++, Celeb-DF v2, DeepFake Detection, Deepfake Detection Challenge, and FFIW10K-v1 datasets for generously providing open datasets that support our research.

### Funding

The authors received no funding for this work.

### Competing Interests

The authors declare there are no competing interests.

### Author Contributions

- ZhiYong Tian conceived and designed the experiments, performed the experiments, analyzed the data, performed the computation work, prepared figures and/or tables, and approved the final draft.
- Junkai Yi conceived and designed the experiments, performed the experiments, analyzed the data, performed the computation work, authored or reviewed drafts of the article, and approved the final draft.

### Data Availability

The code is available in the Supplemental File and at Zenodo: tianzhiyong. (2025). Code_GFADE. Zenodo. https://doi.org/10.5281/zenodo.15104245.

The data is available in Zenodo: tianzhiyong. (2024). datasets [Data set]. Zenodo. https://doi.org/10.5281/zenodo.14260412.

The weights is available in Zenodo: tianzhiyong. (2024). weights [Data set]. Zenodo. https://doi.org/10.5281/zenodo.14233887.

Third-party datasets are available at:

1. FaceForensics++ dataset: https://github.com/ondyari/FaceForensics

2. Celeb-DF v2 dataset: https://github.com/yuezunli/celeb-deepfakeforensics

3. DeepFake Detection dataset: https://github.com/ondyari/FaceForensics

4. Deepfake Detection Challenge dataset: https://www.kaggle.com/competitions/deepfake-detection-challenge/data

5. FFIW10K-v1 dataset: https://github.com/tfzhou/FFIW.

## Supplemental Information

Supplemental information for this article can be found online at http://dx.doi.org/10.7717/peerj-cs.2879#supplemental-information.

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
