# Peer review of "GFADE: generalized feature adaptation and discrimination enhancement for deepfake detection"

_PeerJ Computer Science, doi:10.7717/peerj-cs.2879_

## Round 0.1 · original submission · Major Revisions

Please consider the comments from the 2 reviewers, and respond appropriately in your revision

Reviewer 1 ·

Basic reporting

Generally speaking, the English language used in this manuscript is comprehensible. The authors have written a clear succinct introduction, with well written background. The literature review is compactly written which makes it easy to read and understand.
On the references, the literature review section is well-referenced and is relevant to the method proposed in the manuscript.
Structure-wise, the manuscript is acceptable and does conform to the format laid out by the Peerj standards.
To summarize, the introduction section does provide a clear motivation for the undertaken work presented in this manuscript.

Experimental design

The content of the article presented in this manuscript is within the aims and scope of the journal. In this manuscript, the authors have proposed an efficient approach to detect deepfake images or videos. A thorough investigation and rigorous experimentations were carried out which include cross-domain tests and ablation studies.
As for the proposed method, there are two major works introduced and these are 1) the inclusion of the mixstyle approach, developed by Zhou et al.[1], and 2) cascading of Arcface loss and Focal loss together with the Cross-entropy loss. Arcface loss was developed by Deng et al.[2] while Focal loss was developed by Lin [3]. Concerning the mixstyle approach, the given explanation is too simplified compared to the original mixstyle work. The original mixstyle work created mini-batches within the batch for all instances and then randomly reshuffled these mini-batches prior to pairing two instances (applied to all instances). Also, the mixstyle technique is applied only during the training phase not during the inference phase. These details unfortunately were not mentioned in the submitted article. Likewise, in Figure 1 the process flow for mixstyle is not indicated.
Apart from the above comment, the authors have provided sufficient discussion from the pre-processing stage to the classification result. The performance metrics used were also clearly explained.
For the references, there is one which was wrongly cited, which is Zhou et al.[1]. Upon checking, this article was not published in the Proceedings of the IEEE Conference on Computer Vision and Pattern Recognition (CVPR), but rather it was published in the International Conference on Learning Representations (ICLR 2021).

References
[1]. Zhou, K., Yang, Y., Qiao, Y., and Xiang, T., “Domain Generalization with Mixstyle”, International Conference on Learning Representations (ICLR 2021).

[2]. Deng, J., Guo, J., Xue, N., and Zafeiriou, S., “Arcface: Additive angular margin loss for deep face recognition. In Proceedings of the IEEE/CVF Conference on Computer Vision and Pattern Recognition, pages 4690–4699.

[3]. Lin, T., “Focal Loss for Dense Object Detection”, arXiv preprint arXiv: 1708.02002.

Validity of the findings

The authors have presented results that show the performance of the proposed method. Detailed comparative analyses with other existing techniques were also conducted. Cross-domain experiments were also and verified the superiority of the proposed technique. The ablation study was also performed to justify the contribution of each loss component, and the finding confirmed the importance and effectiveness of the inclusion of the focal loss and the arcface loss. The conclusion part clearly emphasized the achievement of the proposed method in tackling and solving the undertaken issues. Subsequently, the given future works provide the way forward for the possible extension of the proposed method.

Additional comments

None

Reviewer 2 ·

Basic reporting

This manuscript presents a deepfake detection framework designed to enhance cross-dataset and cross-manipulation generalizability. While existing deepfake detection methods perform well within a single dataset, they often struggle in diverse scenarios due to dataset-specific biases. The proposed approach integrates multiple loss functions to improve feature discrimination and mitigate data imbalance. Additionally, the MixStyle technique is incorporated to introduce diverse visual styles during training, further enhancing the model’s adaptability. Experimental evaluations demonstrate that the proposed method significantly outperforms existing approaches in generalization capability. However, some major issues have been identified as below.

Experimental design

1. The motivations of using arcface and focal losses remain ambiguous. The authors claim that they use focal loss to assign more weights to hard-to-classify samples. Does hard-to-classify sample mean real face images? As classifying real faces is more challenging than fake ones. Besides, due to the binary classification nature of the task of deepfake detection, it is unclear why the authors use arcface loss to enhance the model’s generalizability.
2. The novelty of the Mixstyle module is limited. The style mixture module for deepfake detection has been used in ref h. The main differences should be clarified to enhance the clarity.
3. This work only uses AUC to evaluate deepfake detectors’ effectiveness. More metrics such as ACC, EER, and AP are expected to better interpret the results.
4. The setting details of cross-manipulation evaluation is unclear. Do the authors train the model on three manipulations and test it on the other one? The results in Table 3 seem much higher than the results reported in previous papers.

Validity of the findings

5. While the authors combine three loss functions to supervise the training process. The weights of different loss components need to be further studied in the ablation study.

Additional comments

6. Some key references regarding deepfake detection should be included in this manuscript:
a. Detect and locate: Exposing face manipulation by semantic-and noise-level telltales
b. Beyond the prior forgery knowledge: Mining critical clues for general face forgery detection
c. Digital and physical face attacks: Reviewing and one step further
d. Enhancing general face forgery detection via vision transformer with low-rank adaptation
e. Pixel-inconsistency modeling for image manipulation localization
f. Moe-ffd: Mixture of experts for generalized and parameter-efficient face forgery detection
g. Forgery-aware Adaptive Learning with Vision Transformer for Generalized Face Forgery Detection
h. Open-Set Deepfake Detection: A Parameter-Efficient Adaptation Method with Forgery Style Mixture

---

## Round 0.2 · accepted · Accept

The reviewers assessed the article and confirmed all their raised issues were addressed. I therefore can recommend this article for acceptance.

Reviewer 1 ·

Basic reporting

This article is the revised version of the first submission. As commented before, the article is well-structured and the authors have taken the necessary steps and effort to revise the manuscripts accordingly.

Experimental design

All suggested corrections have been addressed and hence there is no further issue.

Validity of the findings

All suggested corrections have been addressed and hence there is no further issue.

Additional comments

Nil

Reviewer 2 ·

Basic reporting

The authors have addressed all my concerns in the response. So I recommend acceptance of the paper.

Experimental design

The experimental design is reasonable, and the comparisons are fair.

Validity of the findings

The findings are verified by solid experimental results.